# A GPU-Accelerated Method for 3D Nonlinear Kelvin Ship Wake Patterns Simulation

**Xiaofeng Sun** [1,*] **, Miaoyu Cai** [2] **and Junchen Ding** [1]

1   Navigation College, Dalian Maritime University, No. 1 Linghai Road, Dalian 116026, China; djc@dlmu.edu.cn
2   School of Naval Architecture, Ocean & Civil Engineering, Shanghai Jiao Tong University,
    No. 800 Dongchuan Road, Shanghai 200240, China; caimiaoyu@outlook.com
*   Correspondence: xfsun_dlmu@163.com

**Abstract:** The study of ship waves is important for ship detection, coastal erosion and wave drag. This paper proposed a highly paralleled numerical computation method for efficiently simulating three-dimensional nonlinear kelvin waves. First, a numerical model for nonlinear ship waves is established based on potential flow theory, the Jacobian-free Newton–Krylov (JFNK) method and the boundary integral method. To reduce the amount of data stored in the JFNK method and improve the computational efficiency, a banded preconditioner method is then developed by formulating the optimal bandwidth selection rule. After that, a Graphics Process Unit (GPU)-based parallel computing framework is designed, and we used the Compute Unified Device Architecture (CUDA) language to develop a GPU solution. Finally, numerical simulations of 3D nonlinear ship waves under multiple scales are performed by using the GPU and CPU solvers. Simulation results show that the proposed GPU solver is more efficient than the CPU solver with the same accuracy. More than 66% GPU memory can be saved, and the computational speed can be accelerated up to 20 times. Hence, the computation time for Kelvin ship waves simulation can be significantly reduced by applying the GPU parallel numerical scheme, which lays a solid foundation for practical ocean engineering.

**Keywords:** kelvin wake pattern; GPU acceleration; boundary integral method; JFNK method; banded preconditioner method





## 1. Introduction

The focus of this research is on the innovation of highly parallel algorithms to simulate the contours of a three-dimensional free surface that appears to be stationary at the stern of a moving vessel, which are known as "Kelvin ship waves" [1]. Research on the kelvin wave shape has been continuously put to practical use in hull design, ship detection and environmentally friendly shipping policies [2].

Froude [3], a famous naval architect, first comprehensively described the morphology and main characteristics of ship waves. Under the assumption of infinite water depth, Kelvin [4] replaced a moving ship with a pressure disturbance point moving in a constant velocity straight line on the water surface and proposed the famous Kelvin angle of $19.47°$. In recent years, with the further study of ship wave characteristics, Rabaud [5] noted that the wake angle will be less than the well-known Kelvin angle if the vessel speed is sufficiently large. Subsequently, various effect factors for the Kelvin wake form were discussed in plenty of papers, e.g., Froude number [6], non-axisymmetric and interference effects, shear current,surface tension, the bottom topography, submergence depth, finite water depth and viscosity [7], etc. Accordingly, the research method of ship waves has gradually shifted from the previous analytical algorithms to numerical simulation.

The overwhelming majority of analytical algorithms of ship wave patterns concerns linear theories. Havelock [8] provided a linear solution for the problem of flow under a pressure distribution. Such ideal perturbations can also be replaced by a single submerged point source singularity [8] and submerged bodies [2]. Moreover, thin ship theory was also

used in the study of the ship wave pattern [9]. Alongside the development of computer technology, numerical simulation methods are becoming increasing popular, and the research focus has shifted from linear problems to nonlinear problems. Nowadays, there are three numerical methods widely used to solve surface wave problems, including the boundary integral method, finite-difference method and finite-element method. In particular, Forbes [10] apply the boundary integral method to build a series of integro-differential equations, and the full nonlinear free surface flow problem was solved with moderate efficiencies. In more recent times, according to this method, many papers solve fully 3D nonlinear ship waves with meshes between $60 \times 20$ and $181 \times 61$ [11,12]. And Pethiyagoda [6] noted that the points used along the $x$-direction should be more than 100 to make a sufficient standard regarding grid independence.

With increasing mesh size, however, the computation time increases exponentially using only Central Processing Unit (CPU) computation power. Alongside the rapid improvement of the electronics industry, the Graphics Processing Unit (GPU) has become another method of acceleration for optimizing the execution of large numbers of threads. Currently, the powerful GPU parallel computing ability has been used to improve the studies on ocean engineering. Crespo [13] introduced the GPU acceleration technique into the Smoothed Particle Hydrodynamics (SPH) method to simulate complex free-surface flows, showing the high efficiency and stability of the GPU program in the SPH method. Hori [14] simulated 2D dam-break flow by developing a GPU-based MPS code and achieved seven-fold speedup. As for a 3D nonlinear free surface problem, Pethiyagoda [15] combined the GPU acceleration technique with the boundary integral method, and LU [16] developed a GPU-accelerated high-order spectral solver. Xie [17] developed the MPSGPU-SJTU solver with a GPU acceleration technique for the liquid sloshing simulation.

This paper presents a parallel solution framework based on GPU for a nonlinear ship wave problem, in which almost all operations are performed in a GPU device. Since the nonlinear boundary integral equation on each node is independent of the synchronous equations on other nodes, plenty of threads on the GPU can be used to complete the integration operation for each node simultaneously. In addition, the parallel computing method can be used for the calculation of the large-scale linear sparse system, while the complex inversion process is quickly finished by using Compute Unified Device Architecture (CUDA) language. According to this framework, a highly paralleled GPU solver is proposed to simulate 3D nonlinear Kelvin ship waves. The computation speed for the 3D nonlinear ship waves simulation can be significantly increased, which is convenient for studying the larger scale problems. On the other hand, the size of Random-Access Memory limits grid growth, and the application of the banded preconditioner method can greatly save running memory to break through this limitation. The banded preconditioner method helps to achieve the standard for the grid independence.

The rest of the paper is as follows. A brief introduction of the problem formulation is given in Section 2. In Section 3, the banded preconditioner JFNK algorithm is described. In Section 4, the theory and implementation of the GPU acceleration technique are presented. The accuracy, efficiency and capability of the GPU solver are verified in Section 5, and a summary in Section 6 concludes the paper.

## 2. Numerical Model

This paper supposes that a flow is moving at a uniform speed $U$ along the positive x-axis direction. Considering the inviscid incompressible fluid of infinite depth without rotational flow, ignoring the influence of surface tension, the potential flow theory is applied. Therefore, a source singularity of strength $m$ is introduced at a distance $L$ below the surface, as illustrated in Figure 1. The transient waves can be generated with the disturbance of source. The free surface wave height and flow field velocity potential can be expressed as $z = \zeta(x, y)$ and $\Phi(x, y, z)$.

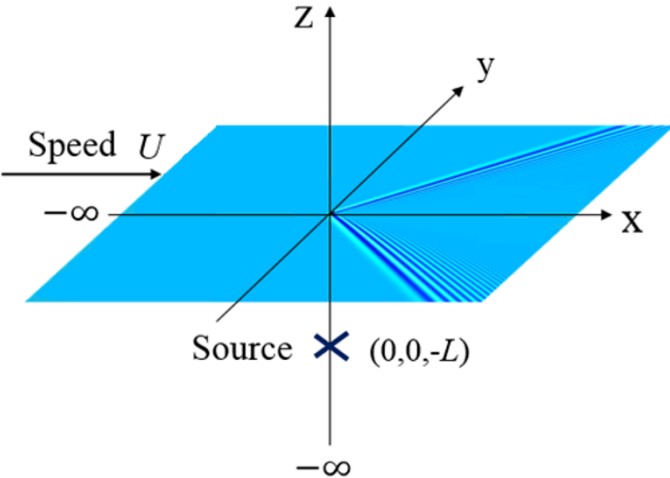

**Figure 1.** Flow field diagram.

Dimensionless analysis is performed with fluid velocity $U$ and distance $L$. The velocity potential $\Phi(x, y, z)$ satisfies Laplace's equation, the free surface kinematic and dynamic boundary condition, the radiation condition and the limiting behavior of source singularity. With $\phi(x, y) = \Phi(x, y, \zeta(x, y))$, the boundary integral equation is written:

$$2\pi(\phi(q) - x) = \int_0^\infty \int_{-\infty}^\infty [\phi(p) - \phi(q) + x - \rho] K_1 d\sigma d\rho$$
$$+ \int_0^\infty \int_{-\infty}^\infty \zeta_\rho(P) K_2 d\sigma d\rho - \frac{\epsilon}{[y^2 + x^2 + (\zeta(q) + 1)^2]^{\frac{1}{2}}} \tag{1}$$

where the $K_1$ and $K_2$ are kernel functions [12].

Moreover, the free surface conditions can be simplified by the symbol $\phi(x, y)$. Then, the kinematic and dynamic boundary conditions of the free surface are combined to be

$$\frac{(1 + \zeta_x^2)\phi_y^2 + \left(1 + \zeta_y^2\right)\phi_x^2 - 2\zeta_x\zeta_y\phi_x\phi_y}{2(1 + \zeta_x^2 + \zeta_y^2)} + \frac{\zeta}{F^2} = \frac{1}{2} \tag{2}$$

To solve the above nonlinear problem numerically, the $N \times M$ mesh is established on the free surface ($N$ and $M$ represent the number of longitude and latitude lines of the mesh, respectively). The $x$-coordinates and $y$-coordinates of nodes are $x_1, x_2, \ldots, x_N$ and $y_1, y_2, \ldots, y_N$ with regular intervals in the coordinate system; thus, the vector $\boldsymbol{u}$ of $2(N+1)M$ unknowns is

$$\boldsymbol{u} = [\phi_{1,1}, (\phi_x)_{1,1}, \ldots, (\phi_x)_{N,1}, \ldots, \phi_{1,M}, (\phi_x)_{1,M}, \ldots, (\phi_x)_{N,M}$$
$$\zeta_{1,1}, (\zeta_x)_{1,1}, \ldots, (\zeta_x)_{N,1}, \ldots, \zeta_{1,M}, (\zeta_x)_{1,M}, \ldots, (\zeta_x)_{N,M}]^T. \tag{3}$$

More $4M$ equations are provided by applying the radiation condition as follows:

$$x_1((\phi_x)_{1,l} - 1) + \gamma(\phi_{1,l} - x_1) = 0 \tag{4}$$
$$x_1((\phi_{xx})_{1,l} - 1) + \gamma((\phi_x)_{1,l} - 1) = 0 \tag{5}$$
$$x_1(\zeta_x)_{1,l} + \gamma\zeta_{1,l} = 0 \tag{6}$$
$$x_1(\zeta_{xx})_{1,l} + \gamma(\zeta_x)_{1,l} = 0 \tag{7}$$

where $\gamma$ is the decay coefficient.

Furthermore, more details about the governing equations, the boundary integral method and numerical discretization are provided by Sun et al. [12].

### 3. Banded Proconditioner JFNK Algorithm

*3.1. Jacobian-Free Newton–Krylov Method*

The JFNK method combines the inexact Newton iteration method with the Krylov subspace method. Its core content is the Generalized Minimum Residual (GMRES) algorithm, according to the matrix free idea, which uses the finite difference form to approximate the product of the coefficient matrix and vector, avoiding the Jacobian matrix calculation and storage alone. The JFNK method mainly has two processes, namely external and internal iterations. The external iteration is the inexact Newton iteration method, and the damping parameter $\lambda_k$ is used to ensure that the nonlinear residual decreases significantly in each iteration for $t = 0, 1, 2, \ldots$ as follows:

$$u_{t+1} = \lambda_k \delta u_t + u_t, \lambda_k \in (0, 1] \tag{8}$$

Its internal iteration is the GMRES algorithm [18], which efficiently solves the correction in inexact Newton iteration; that is, it computes large-scale linear equations as follows:

$$\mathbf{J}(u_t)\delta u_t = -F(u_t) \tag{9}$$

where $\mathbf{J}(u_t) = \partial F(u_t)/\partial u_t$ is the Jacobian matrix [19].

Firstly, the approximate solution of $\delta u_t$ is found by projecting obliquely onto the Krylov subspace

$$K_m(\mathbf{J}_t \mathbf{P}^{-1}, F_t) = \mathrm{span}\{F_t, \mathbf{J}_t \mathbf{P}^{-1} F_t, \ldots, (\mathbf{J}_t \mathbf{P}^{-1})^{m-1} F_t\} \tag{10}$$

where $m$ is the value of the subspace dimension.

An initial linear residual $r_0$ is defined, given an initial guess $u_0$, for the Newton correction,

$$r_0 = -F(u_0) - \mathbf{J}_0 \mathbf{P}^{-1} \delta u_0 \tag{11}$$

Subsequently, $\|r_t\|$ is minimized to a suitable value by the GMRES iteration wherein Jacobian-vector products are approximated with finite difference:

$$\mathbf{J}_t \mathbf{P}^{-1} v \approx \frac{F(u_t + h\mathbf{P}^{-1}v) - F(u_t)}{h} \tag{12}$$

where $v$ represents an arbitrary vector used in building the Krylov subspace [20], and $h$ is a small perturbation

$$h = \frac{\sqrt{(1 + \|u_t\|)h_{mach}}}{\|v\|} \tag{13}$$

Finally, the initial guess $u_0$ can be defined as shown below:

$$\zeta_{1,l} = 0, (\zeta_x)_{k,l} = 0, \phi_{1,l} = x_0, (\phi_x)_{k,l} = 1. \tag{14}$$

The nonlinear equations are solved according to the calculation flow of a banded preconditioner JFNK method, as shown in Figure 2. Note that $v$ in the figure is a unit orthogonal vector in the orthonormal basis of Krylov subspace.

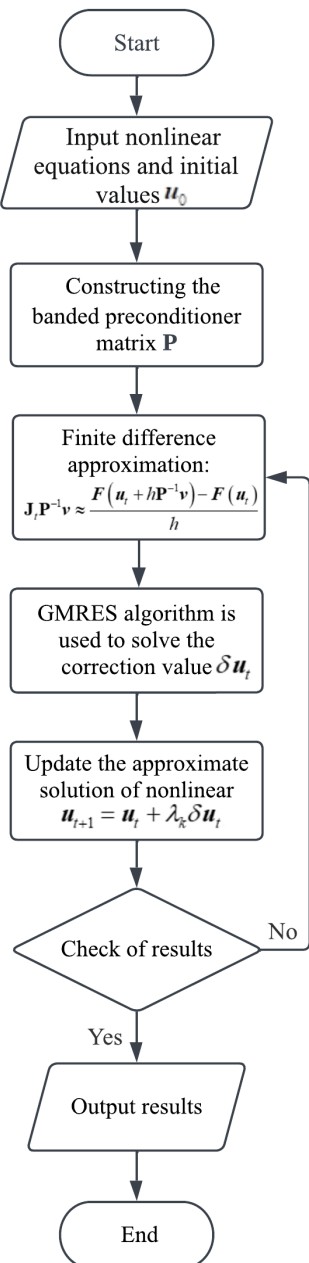

**Figure 2.** Calculation flow chart of the banded preconditioner JFNK method.

*3.2. Banded Preconditioner Method*

Iterative methods, e.g., the GMRES method etc., are currently the most popular choices for solving large sparse linear systems of equations. However, this process of prcconditioning is essential to the most successful application of iterative methods, since the convergence of a matrix iteration depends on the properties of the matrix, e.g., the eigenvalue, etc. [21]. Generally, the methods for choosing the appropriate preconditioner are different for the specific problems. In this section, a banded preconditioner method for solving the nonlinear ship wave problem is proposed.

3.2.1. Building Preconditioner Matrix

For a good preconditioner **P**, it should be cheap to form and to factorize. Meanwhile, the preconditioned Jacobian $\mathbf{J}_t\mathbf{P}^{-1}$ should be easier to solve, which means the eigenvalues are more concentrated. In general, it is feasible to consider a matrix constructed from the

same problem under simplified physics. This paper applies the numerical scheme to the linearized governing equations which apply formally in the limit $\epsilon \to 0$.

The equations of the linear free surface boundary condition are described [22]:

$$\zeta_x = \phi_z \qquad\qquad \text{on } z = \zeta(x, y) \qquad\qquad (15)$$

$$\phi_x - 1 + \frac{\zeta}{F^2} = 0 \qquad\qquad \text{on } z = \zeta(x, y) \qquad\qquad (16)$$

According to the linear free surface boundary condition, the boundary integral equation is described:

$$2\pi(\phi(q) - x) = -\frac{\epsilon}{(x^2 + y^2 + 1)^{\frac{1}{2}}} + \int_0^\infty \int_{-\infty}^\infty \phi_\rho(p) K_3(\rho, \sigma, x, y) d\sigma d\rho \qquad (17)$$

After numerical discretization, the linear system can be described as follows:

$$F1_{k,l} = \phi_{k,l}(q) + \frac{\zeta_{k,l}(q)}{F^2} - 1 \qquad\qquad (18)$$

$$F2_{k,l} = 2\pi(\phi_{k,l}(q) - x_k) + \frac{\epsilon}{[x^2{}_k(q) - y^2{}_l(q) + 1]^{\frac{1}{2}}} - \sum_{i=1}^N \sum_{j=1}^M w(i,j)[(\zeta_\rho)_{i,j} - (\zeta_x)_{i,j}] K^{(3)}_{i,j,k,l} - (\zeta_x)_{i,j} I \qquad (19)$$

where $w(i, j)$ is the weighting function for numerical integration, for $k = 1 \ldots (N - 1)$, $l = 1 \ldots M$. Then, the linear Jacobian can be calculated directly, by differentiating the linear system with respect to $\phi_{1,m}, (\phi_x)_{n,m}, \zeta_{1,m}$ and $(\zeta_x)_{n,m}$. Therefore, the preconditioner matrix **P** can be formed cheaply, and the eigenvalues of $\mathbf{J}_t\mathbf{P}^{-1}$ obviously cluster as shown in Figure 3.

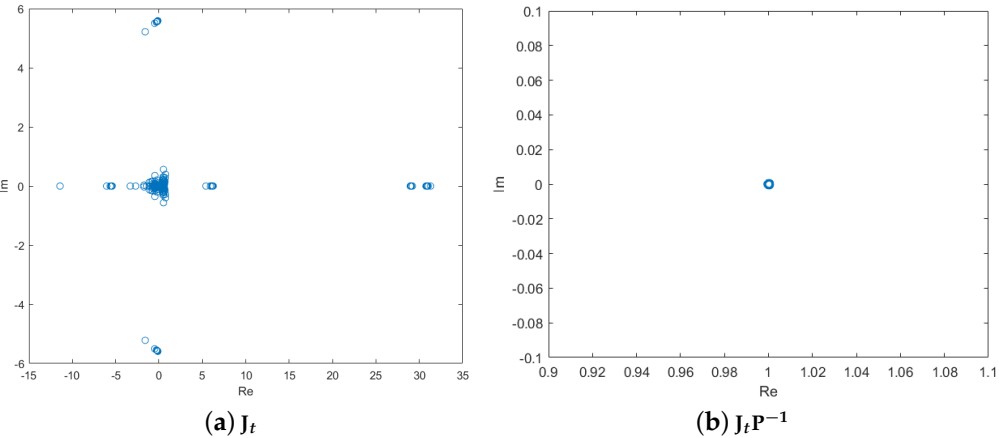

(a) $\mathbf{J}_t$          (b) $\mathbf{J}_t\mathbf{P}^{-1}$

**Figure 3.** The distribution of eigenvalues of $\mathbf{J}_t$ and $\mathbf{J}_t\mathbf{P}^{-1}$ on a $31 \times 11$ mesh.

3.2.2. Preconditioner Factorisation and Storage

The JFNK method requires the result of the product of the inverse preconditioner matrix and vector, $\mathbf{P}^{-1}v$. In general, the operation of inverting a matrix should be converted to solving a system of linear equations, $\mathbf{P}r = v$. Find the solution $r$, and the result of $\mathbf{P}^{-1}v$ will be obtained. In order to calculate this linear system rapidly, the following block matrix method is used to process the preprocessing matrix,

$$\mathbf{P} = \begin{bmatrix} \mathbf{A} & \mathbf{B} \\ \mathbf{C} & \mathbf{D} \end{bmatrix} = \begin{bmatrix} \mathbf{I} & \mathbf{0} \\ \mathbf{CA}^{-1} & \mathbf{I} \end{bmatrix} \begin{bmatrix} \mathbf{A} & \mathbf{0} \\ \mathbf{0} & \mathbf{D} - \mathbf{CA}^{-1}\mathbf{B} \end{bmatrix} \begin{bmatrix} \mathbf{I} & \mathbf{A}^{-1}\mathbf{B} \\ \mathbf{0} & \mathbf{I} \end{bmatrix} \qquad (20)$$

where $\mathbf{I}, \mathbf{A}, \mathbf{B}, \mathbf{C}$ and $\mathbf{D}$ are the unit matrix and the four submatrices.

Accordingly, the vector $v$ can be divided into upper and lower parts $[v_1 \quad v_2]^T$, and then the solution $r$ can be obtained after three cheap steps, as follows:

$$\begin{bmatrix} o_1 \\ o_2 \end{bmatrix} = \begin{bmatrix} v_1 \\ v_2 - \mathbf{C}\mathbf{A}^{-1}v_1 \end{bmatrix} \tag{21}$$

$$\begin{bmatrix} s_1 \\ s_2 \end{bmatrix} = \begin{bmatrix} \mathbf{A}^{-1}o_1 \\ (\mathbf{D} - \mathbf{C}\mathbf{A}^{-1}\mathbf{B})^{-1}o_2 \end{bmatrix} \tag{22}$$

$$\begin{bmatrix} r_1 \\ r_2 \end{bmatrix} = \begin{bmatrix} s_1 - \mathbf{A}^{-1}\mathbf{B}s_2 \\ s_2 \end{bmatrix} \tag{23}$$

The calculation of $\mathbf{P}^{-1}v$ in Equation (12) can be facilitated according to the progressive order from Equations (21)–(23).

### 3.2.3. The Banded Preconditioner

After the factorization operation, the calculation and storage of preconditioner matrix **P** are optimized. However, the size of submatrix **D** is $(N+1)M \times (N+1)M$, and it will increase dramatically as the size of mesh increases. Consequently, there will be two problems when the preconditioner matrix size is large. One is a memory problem, since the running memory of this computer cannot accommodate this preconditioner matrix; the other is an efficiency problem, since inverting the preconditioner matrix will take much time.

By observing the preconditioner matrix, it can be found that the values decay with distance from the main block diagonal. This observation suggests using a banded approximation to the matrix for our preconditioner, as shown in Figure 4. Moreover, batch construction avoids the problem of insufficient running memory due to the large size of the submatrix **D**. The compressed sparse row (CSR) data format is used to save this matrix. Hence, a lot of memory can be saved.

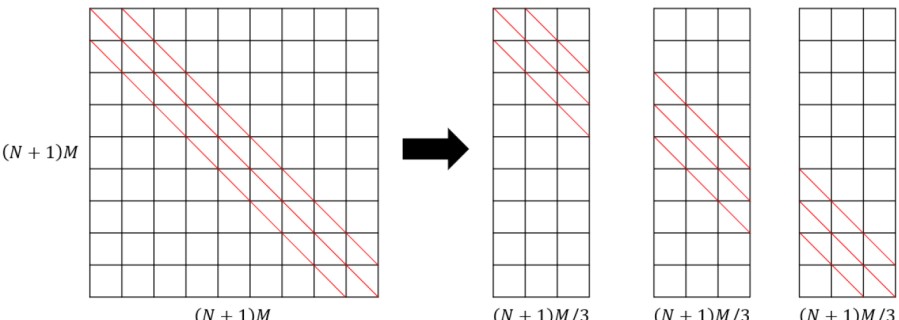

**Figure 4.** Construction of the banded preconditioner, the area marked in red lines indicates the bandwidth.

The feasibility of the banded preconditioner matrix method is verified, as shown in Figure 5. The tightness of clustering can be further improved by increasing the bandwidth. When the $band = 21$, the eigenvalues of $\mathbf{J}_t\mathbf{P}^{-1}$ have been clustered, satisfying the requirement of the GMRES method.

For certain bandwidth values, the computing speed of GMRES will not be significantly improved by increasing the bandwidth further. However, the time required for inverse operation will increase in these cases as the banded preconditioner matrix size increases. The bandwidth regulates the runtime of inverting the banded preconditioner matrix and the number of the inner iterations of the GMRES method. The runtime of inverting the banded preconditioner matrix increases with the bandwidth, while the inner iterations decrease with the bandwidth. Therefore, the total runtime will decrease first and then

increase with the bandwidth, as shown in Figure 6. The case is $\epsilon = 0.4$, $121 \times 41$ mesh and $F = 0.7$, when $b'$ (bandwidth $band = b' \times (N+1)$) is less than 14, an ill-conditioned coefficient matrix is formed, and the accuracy of the solution is low. The runtime decrease with $b'$ ranges from 14 to 16; then, the runtime increases monotonically with $b'$ ranging from 16 to 20. For the case of $121 \times 41$ mesh, the shortest running time is 5.6 s with the optimal bandwidth $band = 16 \times (N+1)$. Therefore, provided that the appropriate bandwidth is selected, it can not only save memory but also improve the computational efficiency.

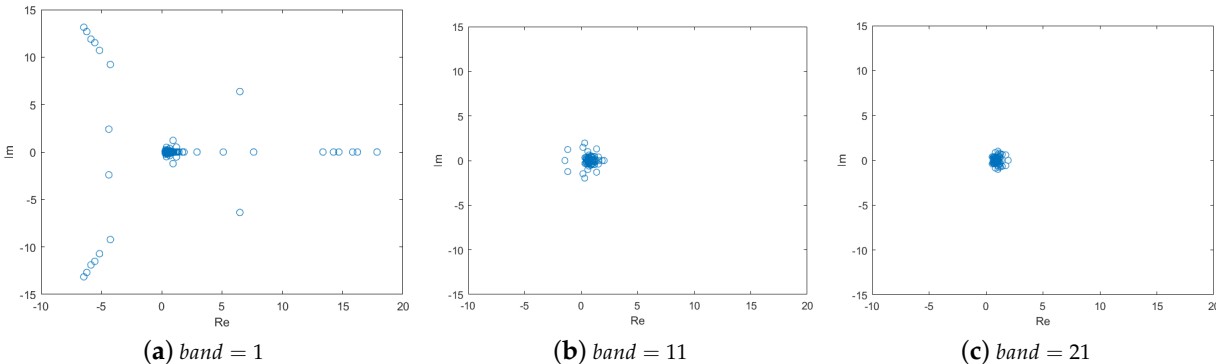

(**a**) $band = 1$      (**b**) $band = 11$      (**c**) $band = 21$

**Figure 5.** The distribution of eigenvalues of $\mathbf{J}_t\mathbf{P}^{-1}$ on a $31 \times 11$ mesh for: $band = 1, band = 11, band = 21$.

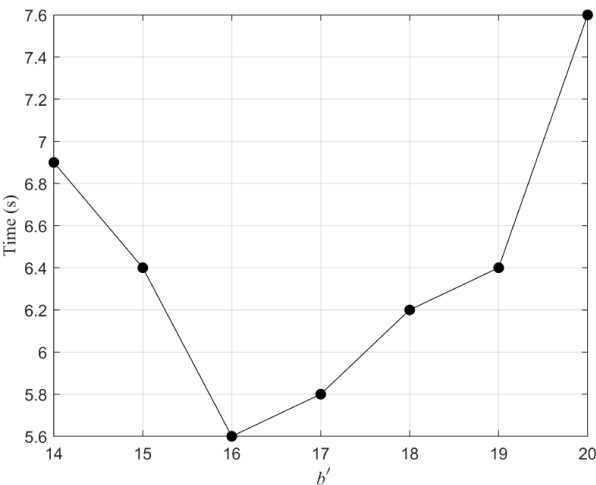

**Figure 6.** The plot of runtime against the bandwidth computed on a $121 \times 41$ mesh, $band = b' \times (N+1)$.

## 4. GPU Parallel Computing Framework

Although the banded preconditioner JFNK algorithm can improve the computational efficiency of the nonlinear ship wave problem, the running time of the program will increase significantly with the increase of the mesh size, which is very unfavorable to the further study of nonlinear ship waves. The reason is that the CPU is not good at handling such large-scale nonlinear equations. Compared with the CPU, the GPU possesses more arithmetic logic units in the same chip area [23]. The computational efficiency of nonlinear ship waves can be greatly improved by utilizing the GPU acceleration technique.

### 4.1. Parallel Computing Framework Design

Compute Unified Device Architecture (CUDA) language is used to develop the numerical scheme for computing ship wave patterns. CUDA is a parallel computer platform and programming model developed by NVIDIA, powered by the GPU [23]. The CUDA toolkit includes abundant GPU-accelerated libraries, tools and a runtime library, which can be compiled in the C language, C++ language and Fortran language. In addition, the CUDA source program can be executed on multiple GPUs. By applying the hybrid programming

model, the parallel computing process consists of a kernel function on the device and serial code on the host CPU. Figure 7 shows the CUDA execution mode and thread hierarchy.

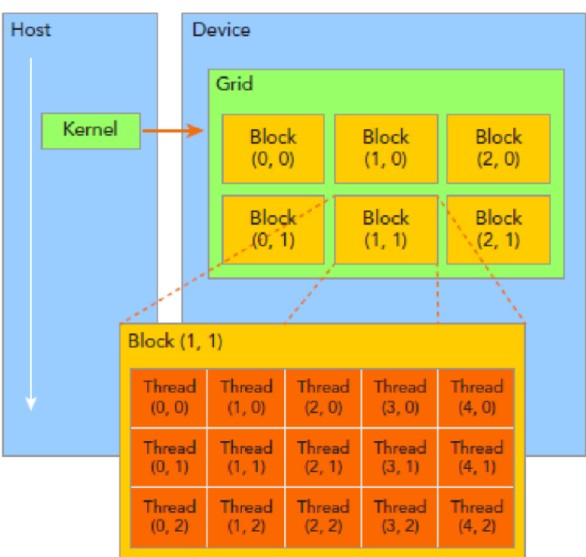

**Figure 7.** Illustration of CUDA execution mode and thread organization hierarchy [23].

As for the solver of ship wave patterns, as described above, there are four main parts: building a preconditioner matrix, creating a nonlinear system, inverting a preconditioner matrix and solving linear equations by the GMRES algorithm. Simulation results of the CPU solver proposed by Sun et al. [12] show that creating a nonlinear system and inverting a preconditioner matrix take up most of the time, as shown in Figure 8. This figure shows the computation time distribution of the CPU solver on a $151 \times 51$ mesh case. The total runtime is 185.8 s, in which the runtime of the inverting preconditioner matrix and time requires for the creation of a nonlinear system is 95.8 and 80.4 s, respectively. Each of them takes up nearly half of the total runtime. Therefore, calculations on these two parts parallelly are vital for improving computational efficiency. And building the preconditioner matrix and solving linear equations will also be executed in a GPU to further shorten the program running time.

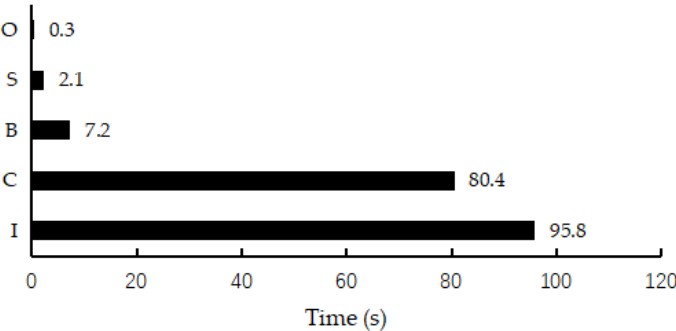

**Figure 8.** The computation time distribution of a ship wave solver. The alphabet I represents the step of inverting the preconditioner matrix, the alphabet C represents the step of creating a nonlinear system, the alphabet B represents the step of building a preconditioner matrix, the alphabet S represents the step of solving linear equations by the GMRES algorithm and the alphabet O represents the step of other codes in the solver.

Based on the above analysis, the GPU solver of Kelvin ship waves adopts a hybrid programming model. The entire parallel computing procedure is shown as follows:

Step 1: Input calculation parameters including the initial guess; the data are transferred from the CPU to GPU;

Step 2: According to the calculation parameters, the nonlinear equations are created in the GPU device;

Step 3: The banded preconditioner method is applied to build the banded preconditioner matrix in the GPU;

Step 4: The QR decomposition algorithm is used to invert the preconditioner matrix, and it saves the decomposition results outside the loop body to avoid repeated QR decomposition of preprocessing;

Step 5: The result of $\mathbf{P}^{-1}v$ is calculated directly using the QR decomposition results; by combining the result of $\mathbf{P}^{-1}v$ with the approximate solution $u$ of the nonlinear equations, the finite difference approximation is carried out to obtain the linear equations;

Step 6: The GMRES algorithm is used to calculate the linear equations, obtain the correction values and update the approximate solutions $u$;

Step 7: Check the approximate solutions of the nonlinear equations: if the accuracy requirement is not met, back to step 5; if the accuracy requirement is met, the result is transferred from the GPU to the CPU.

The corresponding calculation flow chart is shown in Figure 9, which shows the calculation procedure more clearly.

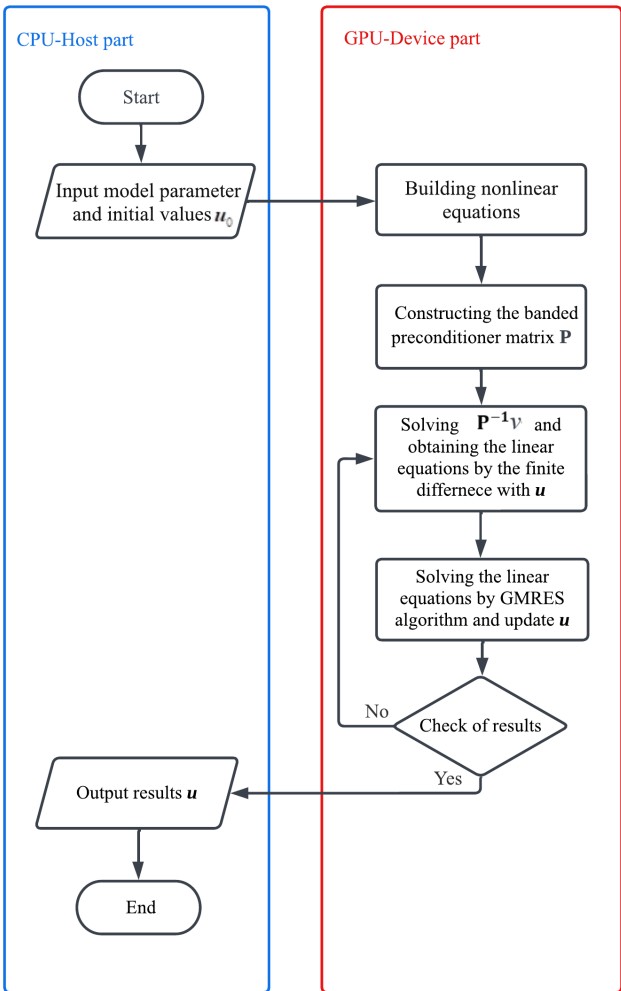

**Figure 9.** The computational flow chart of GPU implementation.

### 4.2. GPU Solver Implementation

4.2.1. Creating Nonlinear System

The programming for creating a nonlinear system on a GPU by using CUDA language is briefly shown in Algorithm 1. On the whole, the dimension of the GPU grid is the equivalent of the size of the mesh, which means that one block can complete the relevant equations of one node in the mesh. One block has 1024 threads; these threads can calculate Equation (1), Equation (2) and Equations (4)–(7) simultaneously.

---

**Algorithm 1** The programming of creating nonlinear system on the GPU

---

**Require:** Initial values and model parameters
**Ensure:** Nonlinear System
  **for** Device Part **do**
    **for** __device__double Integral(double a,double b,double c,double d,double e) **do**
      double val= b/sqrt(c)*log(2*c*a + d*b+ 2*sqrt(c*(c*a*a+ d*b*b +e*b*b)));
    **end for**
    **for** __device__double Integrall(double a,double b,double c,double d,double e) **do**
      double val = a/sqrt(e)*log(2*d*b + d*s+ 2*sqrt(e*(c*a*a+ d*a*b +e*b*b)));
    **end for**
    threadPos = threadIdx.x; k = blockIdx.x; l = blockIdx.y ;
    **for** $threadPos = 0$ to $M \times N$ **do**
      Calculate necessary values
    **end for**
    **for** $threadIdx.x = blockDim.x - 1$ to $blockDim.x - 16$ **do**
      Calculation of the 16 parts to the closed integral
    **end for**
    **for** $i = blocDim/2$ to 0 **do**
      Sum up all thread contributions
    **end for**
    **if** $(k == 0\&\&l == 0 \ldots k == 0\&\&l == 5)$ **then**
      Split the free surface condition and radiation conditions between 5 blocks
    **end if**
  **end for**
  **for** Host Part **do**
    InitialData(double* cpuData, double* gpuData);
    cudaMemcpy(cpuData, gpuData, Datasize, cudaMemcpyHostToDevice);
    dim3 block(1024, 1); dim3 grid(M, N - 1);
    Nonlinear≪ grid, block≫(gpuData);
    cudaMemcpy(gpuData, cpuData, Datasize, cudaMemcpyDeviceToHost);
  **end for**
  **return** Nonlinear System

---

In the device part, there are two device functions which are utilized by the kernel function multiple times. These two device functions are formed to solve the singularity in the second integral of the boundary integral equation. The 16 special threads are set separately for fast computation. Other threads with the same CUDA code are used to complete the calculation of the remaining parts of the boundary integral equation. After the threads have finished computing, all thread contributions are summed up, and $(N-1)M$ nonlinear equations are built. Then, we arbitrarily choose five blocks to calculate the free surface condition and the radiation condition, after which $(N+3)M$ nonlinear equations can be obtained. Therefore, these equations are formed by using the $1024 \times (N-1)M$ threads on the GPU.

In the host part, the environment variables are configured firstly. Then, the data are transferred from the CPU to GPU, and the parallel instruction is sent to the GPU. Finally, the CPU obtains the computation results from the GPU.

4.2.2. Building Preconditioner Matrix

The building preconditioner matrix is decomposed to several tasks that can be operated in parallel by corresponding kernel functions in GPU blocks. The parallel idea and program structure are roughly similar to the part of creating nonlinear equations.

In the device part, in order to avoid data storage conflicts in the GPU, three kernel functions are used to construct the preconditioner matrix in turn. As mentioned above, the preconditioner matrix size is $2(N+1)M \times 2(N+1)M$, and four submatrices are formed by the block decomposition method and banded preconditioner method: submatrix **A** is a tridiagonal matrix of size $3 \times (N+1)M$; submatrix **B** and **C** only differ between coefficients, and the base matrix $\mathbf{B_0}$ can be constructed to represent them, respectively, with a size of $(N+1) \times (N+1)$, $\mathbf{B} = 1/F^2 \cdot \mathbf{B_0}$ and $\mathbf{C} = 2\pi \cdot \mathbf{B_0}$; the submatrix **D** is a sparse matrix with a size of $band \times (N+1)M/3$. Firstly, $M \times (N+1)$ thread blocks are called in the GPU to fulfill the parallel construction of the four submatrices by the kernel function precondition, and the two device functions mentioned above are also used to eliminate the singularity of the linear boundary integral equation. Then, the kernel function matrix is written to call $N+1$ thread blocks for the parallel computation of $\mathbf{CA^{-1}B}$, which involves solving multiple right-handed linear systems and matrix multiplication. Finally, the subtraction operation between matrices is completed by kernel function Schur(), and $M \times (N+1)$ thread blocks are called to perform a parallel operation of $\mathbf{D} - \mathbf{CA^{-1}B}$. The programming for building a preconditioner matrix on the GPU by using CUDA language is briefly shown in Algorithm 2.

---

**Algorithm 2** The programming of building preconditioner matrix on the GPU

---

**Require:** Model parameters
**Ensure:** Preconditioner matrix
  **for** function precondition **do**
    **for** $threadPos = 0$ to $N + 3$ **do**
      Calculate submatrix **A**
    **end for**
    **if** $blockIdx.x == 0$ **then**
      **for** $threadPos = 0$ to $N + 1$ **do**
        Calculate basis matrix $\mathbf{B_0}$
      **end for**
    **end if**
    **for** $threadPos = 0$ to $N \times M$ **do**
      Calculate submatrix **D**
    **end for**
    **for** $i = blocDim/2$ to $0$ **do**
      Sum up all thread contributions
    **end for**
  **end for**
  **for** function matrix **do**
    **if** $k == 0$ **then**
      **for** $threadPos = 0$ to $N + 1$ **do**
        Calculate $\mathbf{CA^{-1}B}$
      **end for**
    **end if**
  **end for**
  **for** function schur **do**
    **for** $threadPos = 0$ to $N + 1$ **do**
      Calculate $\mathbf{D} - \mathbf{CA^{-1}B}$
    **end for**
  **end for**
  **return** Preconditioner matrix

---

In the host part, the variables are first defined according to calculation parameters. Then, data are transferred from the CPU to the GPU, the dimension of the thread blocks and thread grid is specified, and finally, the kernel functions precondition(), matrix(), and Schur() are successively released. This part of the host side code is similar to the establishment of nonlinear equations and will not be repeated here.

### 4.2.3. Inverting Preconditioner Matrix

Comparing the Math Kernel Library which is famous for the computation of sparse linear algebra, the cuSolverSP library is generally faster for solving sparse linear systems [24]. In this paper, the cuSolverSP library is adopted to invert the preconditioner matrix. The present sparse linear system is special; the right-hand side of the system $v$ changes continuously in the iteration, whereas the left-hand side does not. The characteristic of the sparse linear system suggests using QR factorization to calculate $\mathbf{P}^{-1}v$ [21]. By QR factorization, the sparse matrix is decomposed into an orthogonal matrix and an upper triangular matrix, which are saved in GPU memory and are directly used to solve linear equations in each iteration. Finally, the preconditioner-vector products $\mathbf{P}^{-1}v$ can be obtained.

Step 1: Using CSR data format to save the preconditioner matrix with an appropriate bandwidth;

Step 2: In the analysis stage, cusolverSpXcsrqrAnalysis() function is used to analyze the sparsity of orthogonal matrix and upper triangular matrix in QR decomposition. This process may consume a large amount of memory. If the memory is insufficient to complete the analysis, the program will stop running and return the corresponding error message;

Step 3: In the preparation stage, cusolverSpXcsrqrAnalysis() function is used to select the appropriate computing space to prepare for QR decomposition. Here, two memory blocks are prepared in the GPU: one to store the orthogonal matrix and the upper triangular matrix, and the other to perform QR decomposition;

Step 4: The cusolverSpDcsrqrSetup() function is called to allocate storage space for the orthogonal and upper triangular matrices based on the results of the preparation stage. Then, cusolverSpDcsrqrFactor() function is used to complete the QR decomposition of coefficient matrix outside the cycle;

Step 5: Using the cusolverSpDcsrqrZeroPivot() function checks the singularity of the decomposition results, if the nearly singular the program terminates operation and error is given, return to step 1 to choose the bandwidth again;

Step 6: In the loop body, the cusolverSpDcsrqrSolve() function is repeatedly called, and the solution of linear equations can be obtained directly by using the decomposition results stored in the GPU;

The main CUDA functions are shown in Table 1.

**Table 1.** The list of CUDA functions for QR factorization.

| No. | Function Name | Goal |
| :---: | :---: | :---: |
| 1 | cusolverSpXcsrqrAnalysisHost(); | Analyze structure |
| 2 | cusolverSpDcsrqrBufferInfoHost(); | Set up workspace |
| 3 | cusolverSpDcsrqrSetupHost(); | QR factorization |
| 4 | cusolverSpDcsrqrFactorHost(); | QR factorization |
| 5 | cusolverSpDcsrqrZeroPivotHost(); | Check singular |
| 6 | cusolverSpDcsrqrSolveHost(); | Solve system |

### 4.2.4. Solving Linear Equations by GMRES Algorithm

In the process of solving linear equations, because the matrix free idea is adopted to avoid the storage of the coefficient matrix, there is no product operation of the coefficient matrix and vector in the GMRES algorithm, so the operations that can be parallel in this

part are operations between vectors. Therefore, this paper mainly uses the cuBLAS library to complete the CUDA programming of the GMRES algorithm to solve linear equations.

The cuBlasDdot() function is used to realize the inner product of vectors in the GMRES algorithm; the vector subtraction is calculated using the cublasDaxpy() function; the cublasDnrm2() function is used to calculate the Euclidean norm of the vector; the cublasDscal() function is used to divide vector and scalar. After obtaining the orthonormal basis of Krylov subspace and the upper Hessnberg matrix, the cublasDrotg() function is used to perform Givens rotation transformation on the upper Hessnberg matrix in a GPU device to obtain the upper triangular matrix. Then, the solution of the linear least squares problem in the GMRES algorithm is obtained, and the cublasDspmv() function is used to achieve an orthonormal basis and vector multiplication to obtain the solution of linear equations.

## 5. Numerical Simulations and Discussion

In this section, a numerical simulation of ship waves in multiple cases is carried out using the CPU and GPU solvers, and the simulation results are discussed. The effectiveness of the developed banded preconditioner JFNK method is first verified. Then, comparisons between the proposed GPU solver and the CPU solver regarding accuracy and efficiency are performed. Finally, we verify the capability of the GPU solver by comparing the simulation results with real ship wakes. The parameters of the computing environment are listed in Table 2.

**Table 2.** The computing environment of a high-performance computing cluster.

|                      | CPU                     | GPU               |
|----------------------|-------------------------|-------------------|
| Card                 | Intel Xeon Bronze 3204  | NVIDIA Tesla A100 |
| Memory               | 64 GB                   | 40 GB             |
| Max Cores            | 6 per node              | 6912              |
| Programming language | C++                     | CUDA, C++         |

### 5.1. Verification of the Banded Preconditioner JFNK Method

To reveal the effectiveness of the banded preconditioner method, numerical simulations on different mesh sizes, i.e., $181 \times 61$, $241 \times 81$, $301 \times 101$ and $361 \times 121$ with $\Delta x = 0.3, \Delta y = 0.3$ are carried out.

The overall runtimes against bandwidth on these four mesh sizes are illustrated in Figure 10. An optimal value of bandwidth $b'$ exists for a certain mesh size. Furthermore, the optimal value of $b'$ increases with the mesh size and approximately equals $\frac{M}{3}$, in which $M$ means the number of latitude lines of the mesh. Therefore, the optimal bandwidth can be set to $\frac{M}{3} \times (N + 1)$ to obtain an optimal efficiency.

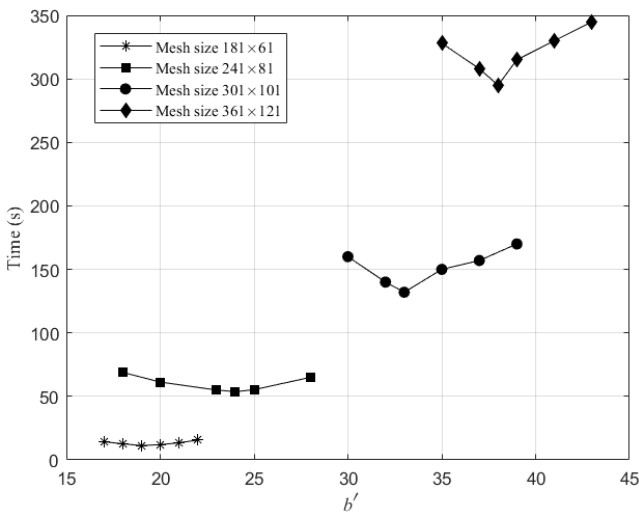

**Figure 10.** Optimal values of bandwidth $b'$ for different mesh sizes.

According to the optimal bandwidth selection rule, the running memories against the bandwidth are shown in Table 3. Correspondingly, the required running memory is drastically reduced by applying the banded preconditioner JFNK method. The mean reduction ratio is about 3.2; this means that the banded preconditioner JFNK method can save running memory by at least two-thirds.

**Table 3.** The running memory usage before and after applying the banded preconditioner method.

| Mesh Size | Before | $b'$ | After | Reduction Ratio |
|---|---|---|---|---|
| $181 \times 61$ | 0.9 1GB | 19 | 0.28 GB | 3.2 |
| $241 \times 81$ | 2.9 GB | 24 | 0.88 GB | 3.3 |
| $301 \times 101$ | 6.9 GB | 33 | 2.3 GB | 3.0 |
| $361 \times 121$ | 15 GB | 38 | 4.6 GB | 3.3 |

*5.2. Verification of the GPU Solver*

5.2.1. Accuracy

To verify the accuracy of the GPU solver, numerical simulations are conducted on $F = 0.7$ and $\epsilon = 0.4$ with a $361 \times 121$ mesh and $\Delta x = 0.3, \Delta y = 0.3$. The simulated wave heights on the centerline are compared with those of the CPU solver proposed by Sun et al. [12], which is shown in Figure 11. Almost all points in the figure are traversed through the center by a line, indicating that the calculation results of the GPU solver are very consistent with those of the CPU solver.

Furthermore, the MSE is used to further explain the error between them, as follows:

$$MSE = \frac{1}{n} \sum_{i=1}^{n} (Truch_i - Value_i)^2 \qquad (24)$$

where $n$ is the amount of data, while $Truch_i$ and $Value_i$ represent CPU results and GPU results, respectively. According to Equation (24), the calculated MSE is $9.37 \times 10^{-8}$, indicating that the calculation error between the GPU and CPU solver is minimal. Since the CPU solver has been verified by Sun et al. [12], the accuracy of the proposed GPU solver can also be acceptable.

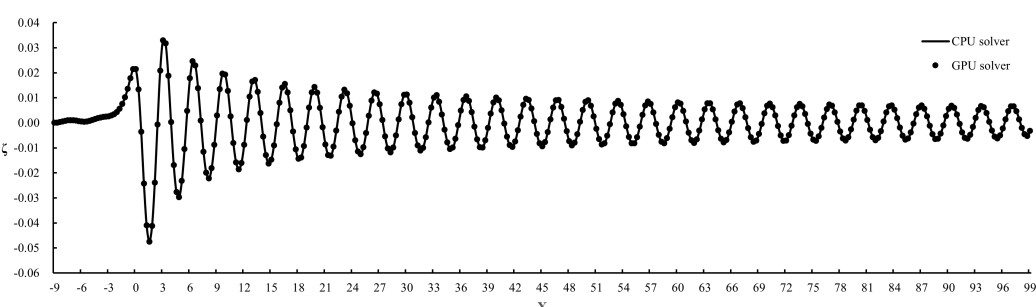

**Figure 11.** A comparison of the centerline profiles for the simulation results of the CPU solver and GPU solver, which are computed on a $361 \times 121$ mesh with $\Delta x = 0.3, \Delta y = 0.3, F = 0.7$ and $\epsilon = 0.4$. The solid line represents the simulation result of the GPU solver, while the solid circles represent the simulation result of the CPU solver.

5.2.2. Efficiency

To verify the efficiency of the GPU solver, numerical simulations are conducted on $F = 0.7$ and $\epsilon = 0.4$ with five mesh sizes, namely $121 \times 41, 181 \times 61, 241 \times 81, 301 \times 101$, $361 \times 121$ and $\Delta x = 0.3, \Delta y = 0.3$. The overall runtimes of the GPU solver are compared with those of the CPU solver proposed by Sun et al. [12] (note that we keep the same calculation parameters as the GPU solver when running the CPU solver), as shown in Figure 12. It clearly shows that the overall runtimes of the GPU solver are much shorter

than those of the CPU solver. The clear accelerated-up ratios between the GPU solver and the CPU solver are shown in Table 4. The accelerated-up ratio on all cases are around 20.0. Therefore, the computation efficiency of the proposed GPU solver is much higher than that of the CPU solver.

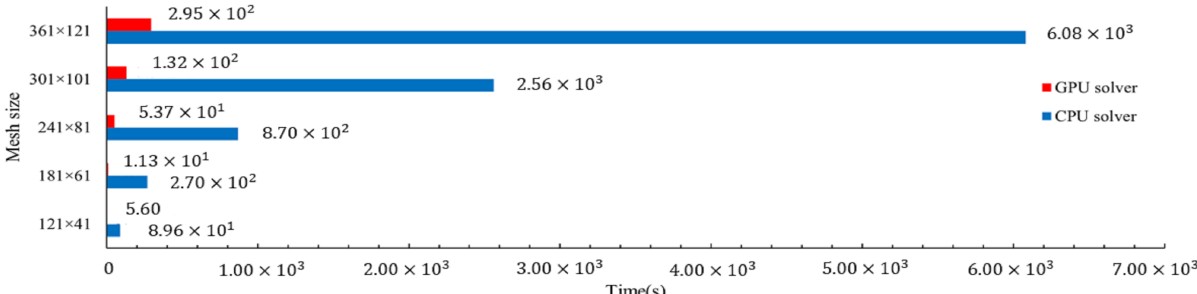

**Figure 12.** The runtime of the GPU solver and CPU solver at different mesh sizes; red bars represent the GPU solver results and blue bars represent the CPU solver results.

**Table 4.** The comparisons of runtime between the CPU solver and GPU solvers on different mesh sizes; the CPU solver is proposed by Sun et al. [12], one GPU solver is proposed in this paper, and the other GPU solver is proposed by Pethiyagoda [15] (results of Exp.).

| Mesh Size | CPU Solver | Exp. | GPU Solver | Accelerated-Up Ratio |
|-----------|------------|------|------------|----------------------|
| $121 \times 41$ | $8.96 \times 10^1$ s | $1.61 \times 10^1$ s | $5.60 \times 10^0$ s | 16.1 |
| $181 \times 61$ | $2.70 \times 10^2$ s | $1.22 \times 10^2$ s | $1.13 \times 10^1$ s | 23.9 |
| $241 \times 81$ | $8.70 \times 10^2$s | $5.51 \times 10^2$ s | $5.37 \times 10^1$ s | 16.0 |
| $301 \times 101$ | $2.56 \times 10^3$ s | $1.78 \times 10^3$ s | $1.32 \times 10^2$ s | 19.3 |
| $361 \times 121$ | $6.08 \times 10^3$ s | $5.04 \times 10^3$ s | $2.95 \times 10^2$ s | 20.6 |

The proposed GPU solver has also been compared with another GPU solver proposed by Pethiyagoda [15] on these cases. The comparison of computation time between them is shown in Table 4. Obviously, the efficiency of the GPU solver proposed in this paper is higher than that of the GPU solver proposed by Pethiyagod [15], and the advantage is more significant with the increase of the mesh size. The reason is that Pethiyagoda [15] only introduced the GPU acceleration technique in the boundary integral method rather than the whole processs, whereas this paper proposes a complete parallel computing framework including the parallel process for inverting the preconditioner matrix. Thanks to the process of inverting the larger mesh size and the heavier computational load, the advantage of the GPU solver proposed in this paper can be more significant.

### 5.2.3. Capability

In the proposed GPU solver, like the CPU solver [12], three parameters can be used to regulate simulation results: namely source strength, source type and Froude number. The wake characteristics can be regulated by adjusting these parameters. For example, the wake waves of vessels of high speed and small overall length can be generated by using the Rankine source, lower source strength and larger Froude number; oppositely, it is appropriate to choose a small Froude number and a higher strength Kelvin source. As shown in Figure 13, the simulation patterns of the GPU solver are compared with the real ship wave patterns. It is found that the simulation patterns are consistent with the real ship waves, while the proposed GPU solver can also generate high-quality simulation patterns for a 3D nonlinear ship wave.

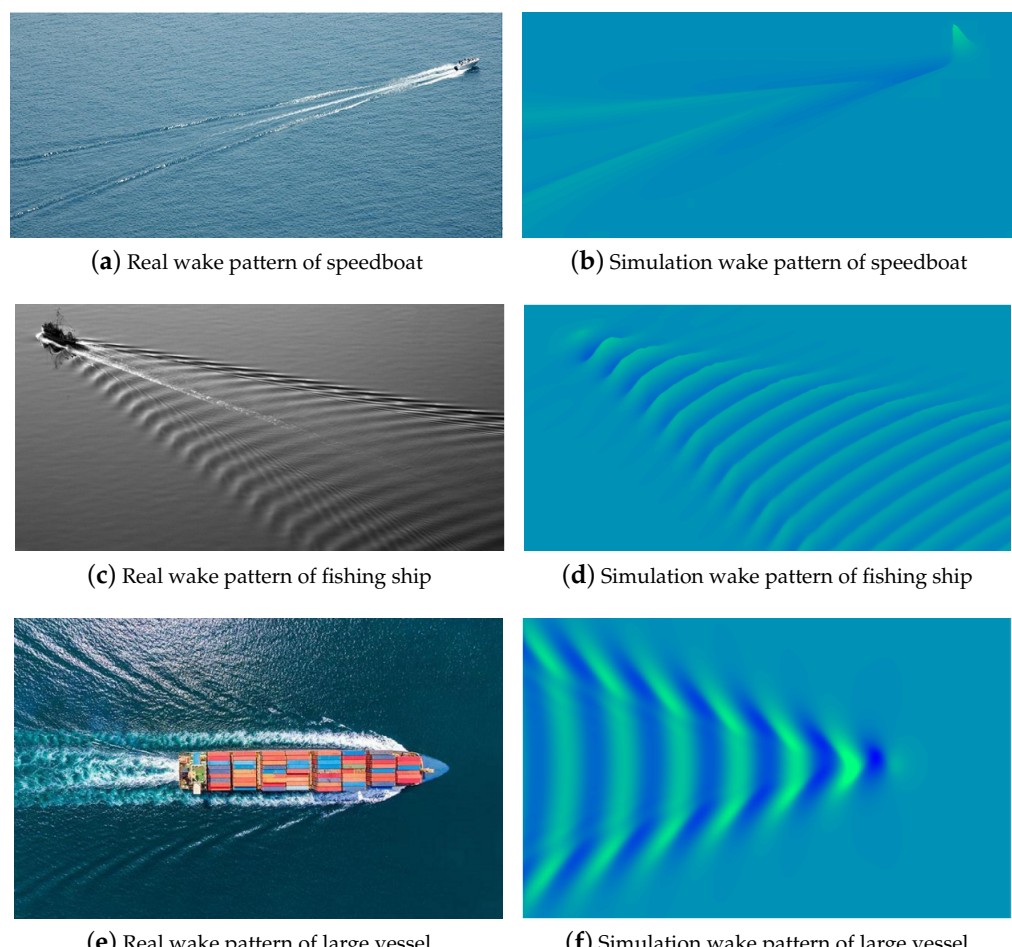

(**a**) Real wake pattern of speedboat     (**b**) Simulation wake pattern of speedboat

(**c**) Real wake pattern of fishing ship     (**d**) Simulation wake pattern of fishing ship

(**e**) Real wake pattern of large vessel     (**f**) Simulation wake pattern of large vessel

**Figure 13.** Wake pattern of real ship waves and the GPU solver computational results. The picture of a real speedboat wake pattern came from the internet https://www.quanjing.com, accessed on 1 September 2023; the picture of a real fishing ship wake came from https://www.shutterstock.com, accessed on 1 September 2023; the picture of a real large vessel wake came from the internet https://blogs.worldbank.org, accessed on 6 September 2023.

## 6. Conclusions

The numerical simulation of ship waves is important for practical ocean engineering. This paper proposes a highly paralleled numerical scheme for simulating three-dimensional (3-D) nonlinear Kelvin ship waves effectively, including a numerical model for nonlinear ship waves, a banded preconditioner JFNK method and a GPU-based parallel computing framework. Numerical simulations show that the proposed GPU solver can save GPU memory and obtain high efficiency significantly. This highly paralleled numerical scheme provides an opportunity for the further study of the nonlinear Kelvin ship waves on a large scale.

(1) The bandwidth has an effect on the running memory and runtime of the GPU solver. Based on the mesh size, the value of the most appropriate bandwidth is around $\frac{M}{3} \times (N+1)$; more than 66% GPU memory can be saved.

(2) The GPU solver can obtain an accurate numerical solution. The mean square error of the GPU solver results and CPU solver results is $MSE = 9.37 \times 10^{-8}$, which is acceptable.

(3) By designing the GPU parallel computing framework, the computation of ship wave simulation is accelerated up to 20 times.

Although a highly paralleled numerical scheme for nonlinear ship waves is proposed in this paper, some assumptions are still made in the construction of the numerical model,

such as infinite water depth and the steady motion of a ship on calm water. It is of great significance to improve simulation results by further exploring the influence of finite water depth, tangential flow and unsteady ship motion on nonlinear ship waves.

**Author Contributions:** Validation, X.S., M.C. and J.D.; investigation, X.S.; writing—original draft preparation, X.S. and M.C.; writing—review and editing, X.S. and M.C.; supervision, X.S. All authors have read and agreed to the published version of the manuscript.

**Funding:** The work was supported by the National Key R&D Program of China (No. 2022YFB4300803, 2022YFB4301402), the Ministry of Industry and Information Technology of the People's Republic of China (No. CBG3N21-3-3), and the National Science Foundation of Liaoning Province, China (No. 2022-MS-159). The authors would like to express sincere thanks for their support.

**Institutional Review Board Statement:** Not applicable.

**Informed Consent Statement:** Not applicable.

**Data Availability Statement:** The data presented in this study are available on request from the corresponding author. The data are not publicly available due to privacy.

**Conflicts of Interest:** The authors declare no conflict of interest. The funders had no role in the design of the study; in the collection, analyses, or interpretation of data; in the writing of the manuscript; or in the decision to publish the results.

## Abbreviations

| | |
|---|---|
| JFNK | Jacobian-free Newton–Krylov |
| GMRES | Generalized Minimum Residual |
| CUDA | Compute Unified Device Architecture |
| GPU | Graphics Process Unit |
| CPU | Central Processing Unit |
| MSE | Mean Square Error |

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
