# Peer review of "A GPU-Accelerated Method for 3D Nonlinear Kelvin Ship Wake Patterns Simulation"

_applsci, doi:10.3390/app132212148_

Round 1
Reviewer 1 Report
Comments and Suggestions for Authors
The paper “A GPU accelerated method for 3-D nonlinear Kelvin ship wave patterns simulation” focuses on the numerical computation method for 3-D Kelvin waves. The numerical methodology carries out parallel computation in GPU-accelerated. The study is interesting and could enrich the literature studies centered on this subject. my suggestion is the comparison of wave patterns not just by visual picture, but also by the angle created by the ship wake has to be measured and the accuracy of the numerical simulation can be verified and validated. Perhaps, using a real ship or model ship the calculated wave pattern can be directly compared with the proposed method.
only minor changes needed to carry out in the text that I found such as line 37...alse...
line 341..computaion....
literature such as MPS and SPH are similar that used GPU for accelerated computation, the authors could add references such as
-Crespo, A.J.C., Domínguez, J.M., Barreiro, A., Gómez-Gesteira, M., Rogers, B.D. GPUs, a New Tool of Acceleration in CFD: Efficiency and Reliability on Smoothed Particle Hydrodynamics. PLoS ONE. 2011, 6(6), 1–13.
-Crespo, A.J.C., Domínguez, J.M., Rogers, B.D., Gómez-Gesteira, M., Longshaw, S., Canelas, R., García-Feal, O. DualSPHysics: Open-Source Parallel CFD Solver Based on Smoothed Particle Hydrodynamics (SPH). Comput Phys Commun. 2015, 187, 204-216.
-Trimulyono, Andi, Haikal Atthariq, Deddy Chrismianto, and Samuel Samuel. "Investigation of sloshing in the prismatic tank with vertical and T-shape baffles." Brodogradnja: Teorija i praksa brodogradnje i pomorske tehnike 73, no. 2 (2022): 43-58.
Reviewer 2 Report
Comments and Suggestions for Authors
See attached
